# Simplified Planner Selection

**Patrick Ferber**[1,2]

patrick.ferber@unibas.ch

University of Basel[1]
Switzerland

Saarland Informatics Campus[2]
Saarland University
Germany

## Abstract

There exists no planning algorithm that outperforms all others. Therefore, it is important to know which algorithm works well on a task. A recently published approach uses either image or graph convolutional neural networks to solve this problem and achieves top performance. Especially the transformation from the task to an image ignores a lot of information. Thus, we would like to know what the network is learning and if this is reasonable. As this is currently not possible, we take one step back. We identify a small set of simple graph features and show that elementary and interpretable machine learning techniques can use those features to outperform the neural network based approach. Furthermore, we evaluate the importance of those features and verify that the performance of our approach is robust to changes in the training and test data.

## 1 Introduction

Planning is concerned with finding a sequence of actions that leads from some initial state to a goal. Over the last decades researchers invented a zoo of different algorithms. All of those exhibiting different strengths and weaknesses. No single algorithm excels on all tasks. To combine the strengths of multiple planning algorithms, in the further course called planners, the idea of having a portfolio of planners to solve a task has emerged. The most common type of portfolios is based on the idea that a planner solves a task either quickly or not at all. Thus, if a planner is not quickly finding a solution, then we could try another planner. Those portfolios posses a set of complementing planners and for each planner they predefined how long the planner runs and in which order they are executed (Helmert et al. 2011; Seipp et al. 2015; Howe et al. 1999). The disadvantage of this approach is that it splits the available time among the planners in its portfolio. It can happen that for some tasks no planners in a portfolio can quickly solve the task. In this case, it would be better to choose the single planner with the highest chance to solve the task and let it run for all the available time. A second portfolio approach has a collection of planners and predicts for a given task how long each planner requires for solving the task or how confident the model is that a planner will solve the task. Then, a single planner is selected and executed. The main obstacle in this approach is finding

a good representation of the task for the predicting model. Fawcett et al. (2014) gathered a large set of handcrafted features. They trained models on those features to predict for some planners how long the planners require to solve a given task. To avoid handcrafting features and potentially ignoring important features Sievers et al. (2019a) translated a given task into a graph which can potentially be translated back into the original task. They interpreted the adjacency matrix of the graph as image, scale the image down to 128x128 pixels, and train a convolutional neural network (CNN) to predict which planner will solve the given task. The idea is that the neural network detects automatically good features. The success of their approach is astonishing. The interpretation of the graph as an 128x128 pixel image ignores a lot of information. Many entries of the adjacency matrix are combined in the same pixel and the information which type a node has is discarded. The success of their CNN shows that the remaining information in the image are sufficient for the planner selection. In a succeeding paper, the input transformation from a graph to an image was eliminated by using graph convolution networks (GCN) (Kipf and Welling 2017) and feeding the graph directly into the neural network (Ma et al. 2020). This caused a modest performance improvement and implies that the images contains sufficient information.

The obvious questions are: What could be the features the neural networks are learning from these graphs resp. images? Can we use those features in combination with simpler machine learning techniques and achieve similar or even better performance? Neural networks are complex function approximators (Cybenko 1989). Today, we are not able to understand which features they have learned. We modify the questions and show that we can find simple features in the graphs such that simple machine learning techniques outperform the neural network based approach. We evaluate how important our features are and verify that our models are robust to changes in the training and test data.

The rest of the paper is structured as follows. Section 2 provides background on planning and on the graph encodings we are working with. Section 3 explains the training setup of our experiments. The experiments and their results are described in section 4 and we summarize our findings in section 5.

## 2 Background

A PDDL task $\Pi^{\text{PDDL}}$ (McDermott et al. 1998) is defined as a tuple $(\mathcal{P}, \mathcal{A}, \Sigma^C, \Sigma^O, I, G)$. $\mathcal{P}$ is a set of first-order predicates. $\mathcal{A}$ is a set of action schemas. $\Sigma^C$ is a set of constant objects, $\Sigma^O$ is a set of non-constant objects. $\Sigma = \Sigma^C \cup \Sigma^O$. An action schema $a \in \mathcal{A}$ with the parameters $p_1, ..., p_n$ is a triple $(pre_a, add_a, del_a)$ with $pre_a \subseteq \mathcal{P}$, $add_a \subseteq \mathcal{P}$, and $del_a \subseteq \mathcal{P}$ and all variables in $pre_a$, $add_a$, and $del_a$ are in $\Sigma^C \cup \{p_1, ..., p_n\}$. $I$ is the initial state and $G$ the goal. Both are sets of atomic statements over $\mathcal{P}$ using $\Sigma$.

Any PDDL task can be translated into a $SAS^+$ task (Bäckström and Nebel 1995). A $SAS^+$ task is a tuple $(\mathcal{V}, \mathcal{A}, s_0, s_*)$ with $\mathcal{V}$ is a set of variables. Each variable $v \in \mathcal{V}$ has a finite domain $dom(v)$. A fact is a pair $(v, v')$ with $v \in \mathcal{V}$ and $v' \in dom(v)$. A *state* assigns every variable $v \in \mathcal{V}$ a value from its domain. A partial assignment assigns every variable in a subset of $\mathcal{V}$ a value from their domains. $\mathcal{A}$ is the set of actions with each action $a \in \mathcal{A}$ is a pair $(pre_a, eff_a)$ and $pre_a, eff_a$ are partial states. $s_0$ is a state which is called the initial state, and $s_*$ is a partial assignment which is called the goal.

We use the previous formalisms to define graph encodings that describe a given task. The problem description graph (PDG) (Pochter, Zohar, and Rosenschein 2011) is an undirected graph which contains for every fact and variable one node and for every action two nodes (one representing the preconditions and one representing the effects of the action). For every value in the domain of a variable the associated node is connected to the node of the variable. Every precondition and effect node associated to the same operator are connected. The nodes associated to the facts in the precondition of an action are connected to the precondition node of an action. The same is done for the facts in the effect of an action. Additionally, we add two special nodes, one is connected to all nodes representing facts that are true in the initial state; the other is connected to all nodes representing facts that are true in the goal.

The second encoding is based on the abstract structure graph (ASG) of the task (Sievers et al. 2019b). An abstract structure is defined as either a symbol (specific elements are symbols), as a list of abstract structures, or as a set of abstract structures. For a given PDDL file, each object and variables becomes a symbol, but also further elements of the PDDL task become symbols. The more complex parts of the PDDL are constructed from those symbols. E.g. the fact *on(a, b)* is constructed as a list containing the symbols *on*, *a*, and *b*. The abstract structure of a PDDL file is written as graph by creating a node for every symbol and every abstract structure and adding a directed edge $X \rightarrow Y$ between two nodes *X* and *Y* if X requires for its definition Y. A PDDL description of a planning task can be directly translated into an ASG and the ASG can be translated back into the same PDDL description.

## 3 Training

To make our results comparable Ma et al. (2020), we perform all experiments on the data set published by Ferber et al. (2019) which extends the data produced by Katz et al. (2018). Our code, new data sets, and experimental results are online available(Ferber 2020).

The data set contains tasks from the classical planning tracks of the International Planning Competitions (IPC) until 2018. Additionally, it includes the domains *BRIEFCASE-WORLD*, *FERRY*, and *HANOI* from the IPP benchmark collection (Köhler 1999), the domain *GEDP* (Haslum 2011), domains from the conformant-to-classical planning compilation (T0) (Palacios and Geffner 2009), and the domain *FSC* (Bonet, Palacios, and Geffner 2009).

For each task the runtimes of 29 optimal planners are measured. The measurements were limited to 30 minutes and at most 7744MiB of memory. We restrict ourselves to the subset of 17 planners that Ma et al. (2020) used. Those are 16 Fast Downward (Helmert 2006) configurations. All configurations are using A* search (Hart, Nilsson, and Raphael 1968) and strong stubborn sets (Wehrle and Helmert 2014). Each of the following eight heuristics is used twice, once with structural symmetries pruning (Shleyfman et al. 2015) using DKS (Domshlak, Katz, and Shleyfman 2012) and once with structural symmetries pruning using orbital space search (OSS) (Domshlak, Katz, and Shleyfman 2015): blind heuristic, LM-cut (Helmert and Domshlak 2009), iPDB (Haslum et al. 2007), a zero-one cost partitioning pattern data base (01-PDB) using a genetic algorithm to compute the pattern (Edelkamp 2006), and four Merge & Shrink (M&S) heuristics (Dräger, Finkbeiner, and Podelski 2006; Helmert et al. 2014; Sievers 2017) using bisimulation (BS) (Nissim, Hoffmann, and Helmert 2011), full pruning (Sievers 2017), Θ-combinability (Sievers, Wehrle, and Helmert 2014), partial abstractions (Sievers 2018), DFP (Sievers, Wehrle, and Helmert 2014), and merging based on either *strongly connected components* (SCC) of the causal graph (Sievers, Wehrle, and Helmert 2016), *maximum intermediate abstraction size minimizing* (MIASM) (Fan, Müller, and Holte 2014), or *score-based MIASM* (sbMIASM) (Sievers, Wehrle, and Helmert 2016). The 17th planner is SymBA* (Torralba et al. 2014). Every planner except for 2 M&S configurations use $h^2$ mutex detection (Alcázar and Torralba 2015).

We removed all tasks from the data set that were not solved by any of the selected planners. 2,439 tasks remain; 145 of those tasks belong to the IPC 2018. For each task the data set contains its PDG, called *grounded*, and its encoding as ASG, called *lifted*. Because our machine learning techniques do not work on graphs, we extract the following 21 basic properties from every graph:

- the number of nodes
- the number of edges
- the graph density ($\frac{\#edges}{\#nodes*(\#nodes-1)}$)
- the number of connected components
- the size of the largest connected component
- the minimum, mean, median, and maximum eccentricity of its nodes (the eccentricity of a node is its maximum distance to any other node; the minimum eccentricity of a graph is called radius, the maximum eccentricity is called diameter)

| | LR | | | | | RF | MLP | | Delfi | |
|---|---|---|---|---|---|---|---|---|---|---|
| | 0 | 0.1 | 1 | 2 | 5 | 50 | 3 | 5 | CNN | GNN |
| Binary | 57.0(0.8) | 86.2(0.0) | 82.1(0.0) | 84.8(0.0) | 88.3(0.0) | 69.9(4.3) | 76.6(8.2) | 77.4(8.2) | 73.1 | 80.7 |
| Log | 62.8(0.0) | 67.6(0.0) | **89.0(0.0)** | 80.7(0.0) | 81.4(0.0) | 66.6(2.4) | 64.8(0.0) | 64.2(1.9) | – | – |
| Time | 56.4(0.7) | 55.2(0.0) | 55.2(0.0) | 52.4(0.0) | 55.2(0.0) | 72.1(3.1) | 68.3(4.6) | 67.4(2.0) | – | – |

| | LR | | | | | RF | MLP | | Delfi | |
|---|---|---|---|---|---|---|---|---|---|---|
| | 0 | 0.1 | 1 | 2 | 5 | 50 | 3 | 5 | CNN | GNN |
| Binary | 65.5(0.0) | 66.2(0.0) | 70.3(0.0) | 64.8(0.0) | 61.4(0.0) | 70.9(4.5) | 61.4(0.0) | 61.4(0.0) | 86.9 | 87.6 |
| Log | 58.6(0.0) | 69.7(0.0) | 69.7(0.0) | 69.7(0.0) | 70.3(0.0) | 73.7(3.5) | 65.2(1.0) | 64.8(0.0) | – | – |
| Time | 65.5(0.0) | 74.5(0.0) | 71.0(0.0) | 69.7(0.0) | 70.3(0.0) | **79.6(5.3)** | 67.9(5.9) | 70.3(4.6) | – | – |

Table 1: Mean coverage and standard deviation (in %) on the IPC 2018 tasks which are solved by at least one planner. Linear regression (LR) uses L1 regularization weights from 0 to 5; random forest (RF) have 50 trees; and the multi-layer perceptrons (MLP) have 3 resp. 5 hidden layers. The last column shows the published performance of the image (CNN) resp. graph (GCN) based versions of Delfi on binary labels. Top: Performance on the grounded graphs. Bottom: Performance on the lifted graphs.

- the minimum, mean, median, and maximum degree of its nodes

- the minimum, mean, median, and maximum in-degree of its nodes

- the minimum, mean, median, and maximum out-degree of its nodes

We selected those properties because they are easy to understand and fast to compute. The values of some properties can greatly differ, e. g. the number of nodes in the grounded graphs vary between 6 and 87,000. Thus, we augment our set of features, by adding the logarithm of each property ( given that the property is always non-zero) and by normalizing each property into the range of 0 to 1. Finally, for every graph we obtain a feature vector with 60 elements. For the runtime, we have the same issues as we had for some properties. The scale of the runtime can vary between fractions of a second and up to half an hour. Therefore, we train the models with three different label transformations: With the original runtime, with the logarithm of the runtime, and with the binary information whether a planner was able to solve a task within the resource limits.

We train plain linear regression models (Galton 1886) and models with L1 regularization (Tibshirani 1996). Linear regression learns for every feature (each property and their transformations) a weight. The output is the weighted sum of the features. L1 regularization adds the L1 norm of the weights as penalty to the optimization process. This causes unnecessary large feature weights to decrease and can filter out irrelevant features. The L1 norm can be scaled with a parameter to make the filtering weaker or stronger.

Second, we train random forests (Breiman 2001). Those are ensembles of decision trees. During training, we optimize each decision tree individually. The final output of the random forest is an averaged decision over all its trees.

The last kind of models we train are multi-layer perceptron. Those are simple neural network consisting of multi-

ple layers of neurons. Each layer is densely connected to the next layer. The value for each neuron is the weighted sum of the neurons connected to it (c. f. linear regression). The value of the neuron is modified by a non-linear function (e. g. $ReLU(x) = max(0, x)$) and is forwarded to the next neurons. We use the Adam optimizer (Kingma and Ba 2015) with a learning rate of 0.001 to optimize the weights of the network.

## 4 Experiments

First, we evaluate how good our simple techniques are at choosing a single planner to solve a given task and compare our results to previous work. Then, we investigate which features have been used and how important those features are. Next, we examine which planners were actually chosen by our models, and we end by evaluating whether our techniques are robust to changes in the data.

One of our feature augmentations normalizes the values of the graph properties. The test data was not used for finding the normalization parameters. All training configurations are run 10 times and their mean coverage as well as their standard deviation are reported. The experiments are run with 3 GB of memory on a single core of an Intel Xeon E5-2660 CPU. The linear regression models finished training in at most 13 seconds, the random forest models in at most 48 seconds, and the neural networks in at most 20 minutes.

### Performance on IPC 2018 Tasks

First, we evaluate how good simple machine learning techniques are at choosing a planner to solve a given task. Like Ma et al. (2020) we use the tasks from the IPC 2018 as test data and all other tasks for training. Neither linear regression nor random forests support validation data, thus, we do not use validation data for the multi-layer perceptrons either.

We train 5 linear regression configurations with L1 regularization weights from 0 to 5, a single random forest with 50 trees, and 2 neural network configurations with 3 resp. 5

hidden layers and 30 neurons in each layer. We use the *sigmoid* activation function and the *cross-entropy* loss to train the networks which make binary decisions. For all other networks we use the *ReLU* activation function and the *mean squared error* loss.

Table 1 shows the performance of all models on the features of the grounded (top) and lifted (bottom) graphs. The two simplest baselines are selecting a random planner for each task which has a coverage of 60.6% and selecting always the planner which performed best on the training data which has a coverage of 64.8%. Most of our trained models outperform both of those baselines. This shows that even simple models can learn useful information. In the grounded setting, linear regression outperforms all other techniques if it is trained on binary or logarithmically transformed labels; on the true labels it is not able to learn anything and performs even worse than the random baseline. Notable, linear regression with our features is even outperforming the Delfi baselines. This does not mean that Delfi is approximating our features, but, it shows that even simple machine learning techniques with understandable features obtain top performances.

The lifted setting is more difficult for linear regression. Its performance is worse in general. In this setting our best performing models are random forests, but even those are not able to outperform the Delfi baselines on the lifted graphs. This means the neural networks of Delfi on the lifted graphs are able to exploit some features that we do not know about. It is an advantage of Delfi that the user does not need to define a set of properties.

## Feature Reduction

Now that we have well performing models, the questions arise which features are required by the models and how important are those features? The answers help us to understand which properties of the graphs describe useful information and which properties can be skipped to speed up the predictions.

Linear regression models allow us to easily investigate their learned weights, thus, we will take a look into the best performing linear regression models for grounded and for the lifted graphs. Additionally, we add the best grounded configuration with binary labels and the lifted configuration with logarithmically transformed labels and and L1 weight of 1 to the comparison. We cannot interpret the magnitude of a weight as importance, because the magnitude of our features varies greatly. Instead, for every feature we sum up how often it has been used by the models. For each configuration we have trained 10 models and each model has internally one linear regression model for each of the 17 planners. Thus, the maximum number of times a feature can be used is 170. The more frequently a feature has been used the more beneficial we can expect it to be. Table 2 shows those sums. Our first observation is that many configurations do not use any normalized feature and rarely use a logarithmically scaled feature. Those transformed features are good candidates to be exclude from training to speed up the predictions.

Upon closer inspection we see that the more precise we

| | Grounded | | Lifted | |
| Features | Binary | Log | Log | Time |
|---|---|---|---|---|
| #nodes | 0 | 170 | 170 | 170 |
| #edges | 170 | 170 | 170 | 170 |
| density | 0 | 0 | 0 | 0 |
| #conn. comp. | 0 | 0 | 0 | 170 |
| max(\|conn. comp\|) | 170 | 170 | 170 | 170 |
| radius | 80 | 150 | 170 | 170 |
| mean eccentricity | 50 | 0 | 160 | 170 |
| median eccentricity | 20 | 20 | 40 | 170 |
| diameter | 50 | 20 | 110 | 170 |
| min. degree | 0 | 0 | 0 | 142 |
| mean degree | 0 | 20 | 0 | 169 |
| median degree | 0 | 0 | 0 | 117 |
| max. degree | 110 | 170 | 160 | 170 |
| min. in-degree | 0 | 0 | 0 | 0 |
| mean in-degree | 0 | 0 | 0 | 79 |
| median in-degree | 0 | 0 | 0 | 0 |
| max. in-degree | 40 | 150 | 160 | 170 |
| min. out-degree | 0 | 0 | 0 | 0 |
| mean out-degree | 0 | 0 | 0 | 31 |
| median out-degree | 0 | 0 | 0 | 140 |
| max. out-degree | 130 | 140 | 170 | 170 |
| log(#nodes) | 0 | 0 | 0 | 170 |
| log(#edges) | 0 | 20 | 0 | 140 |
| log(density) | 0 | 50 | 0 | 160 |
| log(#conn. comp.) | 0 | 0 | 0 | 160 |
| log(max(\|conn. comp\|)) | 0 | 140 | 0 | 160 |
| log(radius) | 0 | 0 | 0 | 170 |
| log(mean eccentricity) | 0 | 0 | 0 | 170 |
| log(median eccentricity) | 0 | 0 | 0 | 170 |
| log(diameter) | 0 | 0 | 0 | 170 |
| log(min. degree) | 0 | 0 | 0 | 140 |
| log(mean degree) | 0 | 0 | 0 | 90 |
| log(median degree) | 0 | 0 | 0 | 170 |
| log(max. degree) | 0 | 0 | 0 | 140 |
| log(max. in-degree) | 0 | 0 | 0 | 169 |
| log(mean out-degree) | 0 | 0 | 0 | 50 |
| log(median out-degree) | 0 | 0 | 0 | 140 |
| log(max. out-degree) | 0 | 10 | 140 | 170 |
| norm(#nodes) | 0 | 0 | 0 | 66 |
| norm(#edges) | 0 | 0 | 0 | 170 |
| norm(density) | 0 | 0 | 0 | 170 |
| norm(#conn. comp.) | 0 | 0 | 0 | 120 |
| norm(max(\|conn. comp\|)) | 0 | 0 | 0 | 104 |
| norm(radius) | 0 | 0 | 0 | 152 |
| norm(mean eccentricity) | 0 | 0 | 0 | 102 |
| norm(median eccentricity) | 0 | 0 | 0 | 152 |
| norm(diameter) | 0 | 0 | 0 | 152 |
| norm(min. degree) | 0 | 0 | 0 | 170 |
| norm(mean degree) | 0 | 0 | 0 | 170 |
| norm(median degree) | 0 | 0 | 0 | 170 |
| norm(max. degree) | 0 | 0 | 0 | 110 |
| norm(mean in-degree) | 0 | 0 | 0 | 110 |
| norm(max. in-degree) | 0 | 0 | 0 | 152 |
| norm(mean out-degree) | 0 | 0 | 0 | 118 |
| norm(median out-degree) | 0 | 0 | 0 | 137 |
| norm(max. out-degree) | 0 | 0 | 0 | 10 |

Table 2: Feature usages for linear regression configurations. The lifted log configuration uses the same L1 weight as the grounded log configuration. All other configurations use their best L1 weight. Four unused features are omitted.

| Features | Grounded | Lifted |
|---|---|---|
| #nodes | 1 | A |
| #edges | 2 | A |
| density | 3 | B |
| #conn. comp. | 4 | C |
| max(\|conn. comp\|) | 1 | A |
| radius | 5 | A |
| mean ecc. | 5 | A |
| median ecc. | 5 | A |
| diameter | 5 | A |
| min. deg. | 6 | D |
| mean deg. | 7 | E |
| median deg. | 8 | F |
| max. deg. | 9 | G |
| min. indeg. | 10 | H |
| mean indeg. | 7 | E |
| median indeg. | 11 | I |
| max. indeg. | 12 | G |
| min. outdeg. | 13 | J |
| mean outdeg. | 7 | E |
| median outdeg. | 14 | K |
| max. outdeg. | 9 | G |

Table 3: Groups of features with a high ($> 0.95$) positive or negative Pearson Correlation.

| Feature Group | Grounded | Feature Group | Lifted |
|---|---|---|---|
| 1 | -2.8% | A | 9.0% |
| 2 | -6.2% | B | -0.7% |
| 3 | 0.0% | C | 0.0% |
| 4 | 0.0% | D | 0.0% |
| 5 | 0.0% | E | -5.5% |
| 6 | 0.0% | F | -1.4% |
| 7 | -2.8% | G | -4.8% |
| 8 | 0.0% | H | 0.0% |
| 9 | -18.6% | I | 0.0% |
| 10 | 0.0% | J | 0.0% |
| 11 | 0.0% | K | 0.0% |
| 12 | -4.8% | | |
| 13 | 0.0% | | |
| 14 | 0.0% | | |
| Baseline | 89.0% | | 74.5% |

Table 4: Performance degradation for the best grounded and lifted linear regression configuration, if a group of highly correlated features is removed.

want to predict the runtimes, the more features the linear regression is using. For the prediction of a binary label the models do not use any transformed feature. To predict the logarithm of the runtimes, some transformed features are used. And to predict the actual runtime, almost all features are used. This trend could be seen in multiple configurations and is independent of using the grounded or the lifted encoding.

A final, less obvious observation is that there are some groups of features for which a trained regressor is only selecting some members of each group. This can be especially well seen with the features *radius*, *mean eccentricity*, *median eccentricity*, and *diameter*. Experiments have shown that removing one of those features has close to no impact on the performance. It turned out that some properties of the graph are strongly linearly correlated. We calculated for each pair of features their Pearson correlation and grouped features together which have an absolute Pearson correlation greater than 0.95. Table 3 shows for both encodings which features are grouped together.

To understand how important each feature group is for the final performance, we retrain the models but withhold a single feature group. Table 4 shows how much the performance of an model changes if a feature group is left out. For the grounded graphs, the most important feature is by far the maximum degree in the graph, the second most important feature is the number of edges. For the lifted graphs, the features 'mean degree', 'mean in-degree', and 'mean out-degree' are the most important. But nearly as important are the features 'maximum degree' and 'maximum out-degree'. Some features groups can be removed without any impact on the test performance and removing feature group 'A' even improves the coverage. Ideally, the L1 regularization would assign those features a weight of zero. This might not happen for two reasons. First, the test tasks - from the IPC 2018 - come from a different data distribution than the training tasks. Thus, features useful for the training tasks might not be useful on the test tasks. Secondly, the loss optimized by the linear regression is not the metric we are comparing. The models try to optimize their prediction for every planner, we are only interested in selecting a single planner to solve the task.

**Planner choices**

To better understand how the models obtain peak performance, we examine which planners are chosen. We want to understand whether those models have learned to choose the right planner for a task. The models do not predict planners at random, because their coverage is not close to the random baseline.

Table 5 shows how often a planner was selected by the best grounded and the best lifted linear regression model. For each planner it shows additionally their coverage on all test tasks ($Cov_T$) and their coverage on those tasks for which they were selected ($Cov_C$). For both configurations we see that the models chose their predictions from a subgroup of (mostly) good planners. We have trained 15 linear regression configuration for the grounded and again 15 configurations for the lifted graph encoding. Could it be that by chance

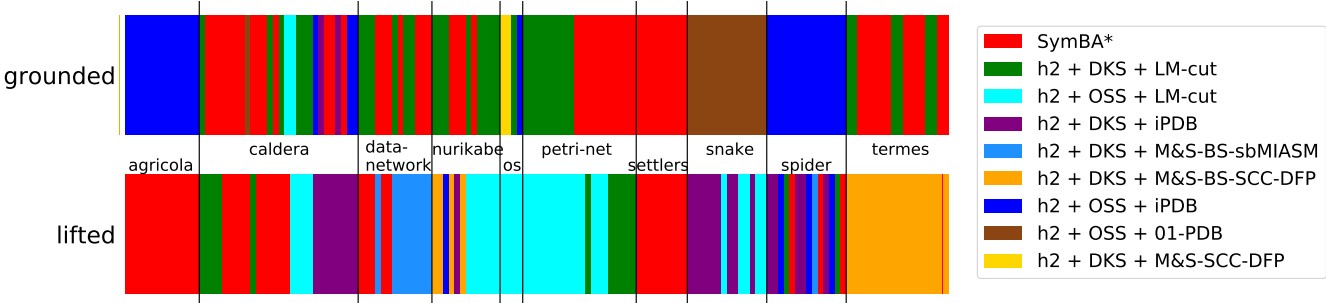

Figure 1: Shows for each task in the test data which planner was selected by the best linear regression configuration for the lifted and grounded graphs. All tasks are sorted by domains and within their domains by name.

| Usage | $Cov_T$ | $Cov_C$ | Planner (grounded) |
|---|---|---|---|
| 39.3% | 82.1% | 84.7% | SymBA* |
| 24.8% | 64.8% | 91.7% | $h^2$ + DKS + LM-cut |
| 21.4% | 70.3% | 80.6% | $h^2$ + OSS + iPDB |
| 10.3% | 59.3% | 80.0% | $h^2$ + OSS + 01-PDB |
| 1.4% | 70.3% | 100.0% | $h^2$ + DKS + iPDB |
| 1.4% | 52.4% | 100.0% | $h^2$ + DKS + M&S-SSC-DFP |
| 1.4% | 64.8% | 50.0% | $h^2$ + OSS + LM-cut |

| Usage | $Cov_T$ | $Cov_C$ | Planner (lifted) |
|---|---|---|---|
| 28.3% | 82.1% | 73.2% | SymBA* |
| 22.5% | 64.8% | 66.3% | $h^2$ + OSS + LM-cut |
| 15.9% | 70.3% | 100.0% | $h^2$ + DKS + iPDB |
| 15.2% | 65.5% | 63.6% | $h^2$ + DKS + M&S-BS-SCC-DFP |
| 9.2% | 64.8% | 62.7% | $h^2$ + DKS + LM-cut |
| 6.2% | 69.7% | 77.8% | $h^2$ + DKS + M&S-BS-sbMIASM |
| 2.8% | 70.3% | 100.0% | $h^2$ + OSS + iPDB |

Table 5: Distribution of how often a planner was selected (Usage), the fraction of tasks, the planner solves from the test tasks ($Cov_B$), and the fraction of tasks the planner solves when being chosen ($Cov_C$) by the best grounded (top) resp. lifted (bottom) linear regression model.

some models found a subgroup of good planners and randomly chooses planners from its subgroup? If this would be the case, then for each selected planner the coverage on the tasks it was selected for should be approximately the same as its coverage on all test tasks. We see that this is clearly not the case. The models have learned to predict for a task which planner is good. Especially in the grounded setting, if the model selects a planner for a task, then the probability of the planner to solve the task is much higher than the planners coverage probability on all test tasks.

A model that has learned which planner is good for a task should assign the same planner to similar tasks. This means, especially within the same domain it should reliably select the same planner. As each domain contains tasks of varying difficulty, it might happen that within the same domain multiple planners are selected, but this should be noticeably different from randomly choosing a planner. Figure 1 shows for every task of the test set which planner was selected. The tasks are grouped by domains and within domains ordered as defined in the IPC 2018. We see that for many domains the models select a single planner. There are also some domains in which the models start with one planner and at one point switch to another planner.

We can conclude that the models have detected some structure in the features of each task and learned to exploit this to select a good subset of planners and to predict a good planner for a task or even for a set of similar tasks.

## Robustness

Until now all experiments have been performed on the same training and test data. One might argue that our results are by chance and with different data the results look different. Thus, we end with two experiments which show that our findings are robust even with changes in the data.

To verify that our performance on the test data does not change significantly with different training data, we split the training data into 10 folds, but enforce that all tasks of the same domain will be assigned to the same fold. Every configuration is trained 10 times, but each time a different fold is ignored. If our approach is robust to changes in the training data, then the performance should not change much. Table 6 shows that indeed the configurations still perform similarly

| | LR | | | | | RF | MLP | |
|---|---|---|---|---|---|---|---|---|
| | 0 | 0.1 | 1 | 2 | 5 | 50 | 3 | 5 |
| Binary | 60.6(5.5) | 81.8(6.1) | 82.0(5.3) | 82.6(5.4) | 84.6(5.7) | 70.4(3.8) | 75.6(7.8) | 73.9(7.9) |
| Log | 63.1(5.7) | 67.2(6.3) | 81.1(8.1) | 78.5(6.4) | 79.8(6.0) | 68.8(5.7) | 64.8(0.2) | 67.0(3.4) |
| Time | 60.0(4.9) | 56.1(5.0) | 56.0(4.8) | 55.5(5.1) | 57.3(5.3) | 71.5(5.9) | 68.7(5.2) | 68.8(5.5) |

| | LR | | | | | RF | MLP | |
|---|---|---|---|---|---|---|---|---|
| | 0 | 0.1 | 1 | 2 | 5 | 50 | 3 | 5 |
| Binary | 63.6(4.0) | 70.3(4.7) | 70.6(3.5) | 70.7(5.2) | 66.1(6.0) | 73.4(5.1) | 63.2(3.7) | 64.7(6.4) |
| Log | 58.5(3.8) | 69.8(3.1) | 68.8(2.9) | 69.6(3.0) | 70.3(1.8) | 74.1(6.0) | 64.8(0.0) | 64.8(0.0) |
| Time | 63.7(3.7) | 73.2(3.8) | 70.3(2.9) | 69.7(3.2) | 70.6(3.6) | 77.4(6.1) | 66.3(2.4) | 69.0(4.1) |

Table 6: Mean coverage and standard deviation (in %) on the IPC 2018 tasks. The training data is split into 10 folds such that all tasks of the same domain are in the same fold. For each experiment repetition a different fold is ignored. Top: Performance on the grounded graphs. Bottom: Performance on the lifted graphs.

| | LR | | | | | RF | MLP | |
|---|---|---|---|---|---|---|---|---|
| | 0 | 0.1 | 1 | 2 | 5 | 50 | 3 | 5 |
| Binary | 85.6(8.3) | 77.3(17.5) | 76.3(17.2) | 76.0(17.0) | 76.6(18.0) | 83.4(6.5) | 76.8(17.2) | 79.7(16.8) |
| Log | 86.7(7.8) | 85.9(8.4) | 82.4(8.5) | 78.2(15.6) | 77.8(16.2) | 83.4(9.5) | 83.8(8.6) | 84.8(6.5) |
| Time | 86.3(8.5) | 84.2(8.5) | 84.3(8.6) | 84.5(9.0) | 84.3(8.8) | 79.2(17.7) | 84.9(4.4) | 83.3(6.5) |

| | LR | | | | | RF | MLP | |
|---|---|---|---|---|---|---|---|---|
| | 0 | 0.1 | 1 | 2 | 5 | 50 | 3 | 5 |
| Binary | 81.5(4.1) | 75.8(15.4) | 75.6(16.5) | 74.1(15.2) | 73.4(15.4) | 77.6(13.3) | 72.7(16.1) | 72.2(16.1) |
| Log | 81.0(9.9) | 73.8(15.5) | 82.3(7.8) | 82.5(7.8) | 82.7(7.8) | 75.9(13.6) | 82.2(8.5) | 82.2(8.5) |
| Time | 82.4(5.6) | 78.7(10.6) | 74.4(15.6) | 74.6(16.8) | 74.4(16.7) | 78.9(13.2) | 81.5(7.7) | 78.9(9.0) |

Table 7: Mean coverage and standard deviation (in %) on a random test fold. The data set - including the tasks from the IPC 2018 - is split into 10 folds such that all tasks of the same domain are in the same fold. For each experiment repetition a different fold is selected as test set. The other folds are used for training. Top: Performance on the grounded graphs. Bottom: Performance on the lifted graphs.

well. The performance might have decreased a bit, because $1/10^{th}$ of the training data was ignored. The standard deviation shows us that depending on which part of the training data is ignored the coverage can moderately vary.

Finally, we also change the test data. We split the whole data set into 10 folds, still keeping tasks from the same domain in the same fold. We train for each configuration 10 models. Each model uses a different fold as test data and trains on the remaining nine folds. Table 7 shows that our coverage increases. This might be because the tasks in the IPC 2018 were quite different from those tasks of previous IPCs, thus, learning from tasks of previous IPCs to select planners for tasks of the IPC 2018 is a difficult challenge. Changing the test data could make the learning easier. Another reason could be that planners in the data set were selected because they are good planners and their goodness could have been measured by using the old tasks of the IPC.

## 5 Summary and Future Work

We have shown that we can use simple machine learning techniques like linear regression to predict for a given tasks which planner to run to solve the task. In the grounded setting this even outperformed the image resp. graph convolution based baselines. Thus, we can have explainable decisions while still keeping top performance. At the same time this is not a justification to forget the image resp. graph convolution approaches. In the lifted setting those perform still better and have the advantage that the user does not need to come up with a set of good features, but the neural networks learn those features themselves.

Additionally to training those models, we studied which features are relevant for the predictions and how important they are. In the grounded setting the maximum degree of the graph was the most important information. On the other hand in the lifted setting there was no single feature with a similarly large impact. Finally, we verified that the models learned which planners to run for a domain.

Some future work is to use more fine grained features of the graph, e.g. number of operator nodes, such that we can reason which properties of the task instead of which property of the graph determine the planner choices.

## Acknowledgments

Patrick Ferber was funded by DFG grant 389792660 as part of TRR 248 (see https://perspicuous-computing.science). This work was supported by the Swiss National Science Foundation (SNSF) as part of the project "Certified Correctness and Guaranteed Performance for Domain-Independent Planning" (CCGP-Plan).

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
