# OpenReview forum: "Simplified Planner Selection"
_icaps-conference.org/ICAPS/2020/Workshop/HSDIP — HSDIP 2020_

### Official Review · AnonReviewer1 · 2020-03-31
**Early assessment**

**Rating:** 8
**Confidence:** 3

**Review:**

The topic of this paper is the selection of algorithms on the basis of features obtained from the problem description graph (abstract structure graph) of a given planning task. For this task, (graph) convolutional neural networks were used in the past. This technique has shown good results, but is difficult to interpret. In other words, it is difficult to infer what a neural network has learned, which is an interesting question to ask in order to infer rules or heuristics when to use a particular planning algorithm. The paper proposes to analyze handcrafted graph features in combination with "traditional" machine learning algorithms. The objective is to determine whether it is possible to achieve the same performance with these features and to perform an importance analysis of the graph features.

The topic of the paper fits to the workshop as the HSDIP workshop explicitly welcomes multidisciplinary work such as the application of machine learning in heuristic search (here: algorithm/heuristic selection).

I agree with the statement that neural networks are complex function approximators, so it is difficult to understand what features have been learned. However, I think this statement would benefit from a citation.The question of whether interpretable machine learning techniques based on a similar input can achieve the same performance as neural networks is certainly interesting and can provide some insight why the PDG and ASG of a planning task seem to be a good tool to determine which planner to use. But, I am not really sure if the graph features used for the elementary techniques of machine learning can really be learned from a (graph) convolutional neural networks. As in the paper mentioned, a lot of information is lost when the graph is converted to a 128x128 pixel image. My first question is whether information is also lost when the graph is used directly as input to the neural network? In cases where information is lost, one can certainly infer whether the graph is dense or not, but I wonder if features such as radius, diameter or degrees are preserved.  This would indicate that the neural network has no chance to use information used by elementary machine learning?

Regarding the planners used, it is not clear whether seq-opt-symba-1 also uses h2 mutexes. To the best of my knowledge, h2 mutexes are used by default in seq-opt-symba-1. It would be good to mention this.

The sentence: "Like Ma et al. (2020), we select the 145 solvable tasks from the IPC 2018 ..." is confusing. I assume that the tasks that were solved by one of the 17 considered planners within the given time and memory constraints are referred to or do only 145 instances from the IPC 2018 have a solution?

Minor comments:

Abstract:
- convolutional resp. graph convolutional neural networks (abbrev. in abstract) -> respectively? -> (graph) convolutional neural networks

Introduction:
 - of the graph as image *image*

Background:
 - eff -> mathit ($\mathit{eff}$)

Experiments:
 - Instead, *for* we sum up for every feature
 - a feature can be used 170 -> a feature can be used is 170
 - the linear regession are using -> is using ?

---

> ### Comment · AnonReviewer1 · 2020-08-17
> **Final remark**
>
> After revisiting the paper, I still think that this is a good paper that fits the workshop, if the points in my early assessment are addressed. Since there is no new version of the paper or response, I have nothing more to add at this time.

---

> > ### Author Response · Authors · 2020-08-18
> > **Thank you for your feedback.**
> >
> > Do you mean by "Since there is no new version of the paper" that there is no new version with significant changes? We uploaded an updated version in June that improves on most of your feedback (thank you again), but does not change the story of the paper or the experiments.
> >
> > The graph representation of the PDDL task does not lose any information and can be translated back to an equivalent PDDL task (the order of the elements in the task, e.g. the actions, might change). The more information is lost, the more unlikely it becomes that we can still infer our properties like radius or degree. In the concrete case of the 128x128 image, you are right, it cannot learn anymore all properties (e.g. degree).
> >
> > seq-opt-symba1 (now called SymBA*) uses h2 mutexes. We added this to the paper.
> >
> > As you assume the 145 test instances of the IPC are those tasks which could be solved by at least one of our planners. We rephrased the section.

---

> > > ### Comment · AnonReviewer1 · 2020-08-18
> > > **Thanks for the answer**
> > >
> > > Oh, openreview did not show any new pdf uploads or revisions, so I assumed that the pdf had not changed. Thank you for pointing that out. Thank you for answering my questions and incorporating my suggestions.
> > >
> > > Questions:
> > > It appears that the values in Table 1, 5 and 6 have changed. The original values seem to be rounded, which I think has been made more precise in the new version?
> > >
> > > Overall:
> > > There has been a significant revision of the paper, which clarifies several things. I have updated my score accordingly and stick to my original assessment that the paper fits well to the workshop.

---

> > > > ### Author Response · Authors · 2020-08-18
> > > > **Next answer**
> > > >
> > > > We updated the values in the old table 1, 5, 6 (new 1, 6, 7). Previously they were rounded to integer values (which was not intended). Now, they are rounded to the first decimal place.

---

### Official Review · AnonReviewer2 · 2020-04-07
**Early assessment**

**Rating:** 7
**Confidence:** 3

**Review:**

The paper builds on a recent line of research on using machine learning for portfolio planner selection. In contrast to prior research that used images and graph representations of the planning problem as input to the neural network, the approach proposed here does feature engineering, and extracts a number of features from the planning graph. These features are fed to different classifiers to be trained on: linear regression, random forests, and multilayer perceptron.

The motivation for the need for good portfolio planners is well articulated. I found that the paper does a good job at presenting the work, particularly to these readers that may be familiar with the planning literature, but not so much with the machine learning literature. I liked the extensive experimental section and the different experiments and tests that were performed and evaluated, and I also liked how the results were interpreted.

The topic of the paper is in the scope of the workshop, and in my opinion the quality of the submitted work deserves being presented at HSDIP.

There are some typos and some parts of the text that need to be rephrased, but in general it is well written. Perhaps for completeness, I recommend the authors to include a sentence of two to explain what is the approach that you follow from Sievers et al. (2019), and that you point out in the beginning of Section 3.

---

> ### Author Response · Authors · 2020-08-18
> **Thank you for your review.**
>
> The updated version already fixes some typos and improves the phrasing. New typos we detect will be fixed for the camera ready.

---

### Comment · Program_Chairs · 2020-09-14
**Final Decision: Accept**

Dear Authors,

Thank you very much for your submission. We are happy to inform you that we have decided to accept it and we look forward to your talk in the workshop. You will receive additional information per mail in the coming days.

Best,
The HSDIP'20 team

---

### Decision · Program_Chairs · 2020-09-30

Accept